# Valorising Insect Exoskeleton Biomass Filler in Bioplastic-Based Eco-Friendly Rigid Items for Agriculture Applications

**DOI:** 10.3390/polym17070943

**Published:** 2025-03-30

**Authors:** Norma Mallegni, Vito Gigante, Steven Verstichel, Marco Sandroni, Neetu Malik, Miriam Cappello, Damiano Rossi, Sara Filippi, Andrea Lazzeri, Maurizia Seggiani, Patrizia Cinelli

**Affiliations:** 1Department of Civil and Industrial Engineering, University of Pisa, Largo Lucio Lazzarino 1, 56122 Pisa, Italy; vito.gigante@unipi.it (V.G.); marco.sandroni@unipi.it (M.S.); neetu.malik@ing.unipi.it (N.M.); miriam.cappello@unipi.it (M.C.); damiano.rossi@unipi.it (D.R.); sara.filippi@unipi.it (S.F.); andrea.lazzeri@unipi.it (A.L.); maurizia.seggiani@unipi.it (M.S.); 2Normec OWS, Pantserschipstraat 163, 9000 Ghent, Belgium; steven.verstichel@normecgroup.com

**Keywords:** biocomposite, insect exoskeleton, compostable, biodegradable, circular economy

## Abstract

In this work, by-products from insect farming valorisation are proposed as filler in biocomposite production, with relevant biodegradation in compost and valuable thermal and mechanical properties. Thus, we report on the preparation, properties, and biodegradability in compost of composites based on Poly(butylene succinate-co-adipate) (PBSA) and Poly(3-hydroxybutyrate-3-hydroxyvalerate) (PHB-HV) (70/30% by weight as a polymeric matrix, with filler from insect exoskeleton (I) up to 15% by weight in the 85% by weight of polymeric matrix. The insect biomass was a by-product obtained from grinding the insect’s post-protein extraction dry exoskeleton. The composites were produced by melt extrusion and characterised in terms of processability, thermal stability, morphology, and mechanical properties to select formulations optimised for injection moulding processing. The optimised composites (PBSA/PHB-HV) with 15% by weight of filler were used to produce pots by injection moulding on an industrial scale extruder. Selected formulations were tested for biodegradability in compost, which evidenced the relevance of insect exoskeleton filler for meeting the requirements for the disintegration of rigid items. This paper presents a sustainable option for valorising the insect exoskeleton residue that remained after protein extraction for animal feed production and reducing the production cost of PBSA/PHB-HV-based composites without compromising the mechanical properties for application as rigid items in agriculture, all while promoting biodegradability in industrial compost.

## 1. Introduction

The continuous growing trend of plastic production generates concerns for the disposal and management of post-consumer plastic, particularly those intended for single-use, hard-to-collect, and hard-to-recycle applications. The global primary plastic production is estimated to reach 1100 million tons by 2050. Around 36% of all plastics produced are used in packaging, including single-use items for food and beverage containers and agricultural use. Unfortunately, approximately 85% of these plastics end up as waste in landfills or unregulated areas. Moreover, almost 98% of single-use plastic products are derived from fossil fuels or “virgin” feedstock. The associated greenhouse gas emissions from the production, usage, and disposal of traditional fossil fuel-based plastics are anticipated to contribute to 19% of the global carbon budget by 2040 [1].

The renewed focus on developing sustainable materials has prompted studies into biodegradable polymers that can replace conventional plastics derived from petroleum. Adipic and succinic acids are directly esterified with 1,4-butanediol to produce Poly(butylene succinate-co-adipate) (PBSA), a thermoplastic aliphatic random co-polyester [2]. Biobased succinic acid is the source of some manufacturers who market partially biobased poly(butylenee succinate) (PBS) and PBSA, which achieve a high biobased content, up to 54 weight percent; this is envisaged to increase shortly due to the availability of biobased building block for PBS and PBSA production [2]. PBS and PBSA are noted for their excellent mechanical properties and processability, like extrusion, film blowing, thermoforming, and injection moulding, mirroring those of polyolefins [1,3,4,5].

PBSA differs from PBS by having higher values for the impact strength, elongation at break, and flexibility, while having a lower crystallinity, tensile strength, and melting temperature [6,7]. Due to its mainly amorphous structure, PBSA has a pretty fast rate of degradation in soil, ocean, industrial composting, and activated sludges [2,5], which makes it especially desirable for different applications, where biodegradability is a relevant property, such as most agriculture applications, including plant pots, bag liners, and agricultural mulching films [7]. Another highly biodegradable family of polymers is that of the polyhydroxyalkanoates (PHAs), which are synthesised by various bacteria strains under stress conditions and have qualities comparable with those of several commodity plastics [8,9,10]. Because of the easier processability and versatility in use, copolymers, such as poly(hydroxybutyrate-co-valerate) (PHBV), have garnered a great deal of attention in both academic and industrial domains [11], making them suitable for the production of items by injection moulding [12,13], with the advantage of being biodegradable in several environments [14,15]. By the way, both PHAs and PBSA have quite high prices (5–8 EUR/kg and 7–12 EUR/kg [10], respectively) compared with other biodegradable polymers, such as Polylactic acid (PLA), which limits their use in common low-cost applications, such as packaging and agriculture devoted items. Moreover, even if the polymers were positively tested for meeting mineralisation in compost according to EN 13432 (2000) “Packaging. Requirements for packaging recoverable through composting and biodegradation. Test scheme and evaluation criteria for the final acceptance of packaging”, rigid items realised with these polymers may not meet the standard for disintegration when the thickness of the items is relevant [16,17]. Indeed, compostability is the main advantage in using these polymeric matrices in agriculture since it allows for disposal with the organic biomass versus the conventional polymers which are not biodegradable and not recyclable due to deterioration after application in the open environment and to the contact with soil and fertilisers, pesticides, etc. In this context, natural biomass, by-products, or waste products with almost no value, or eventually with an associated cost for disposal, suitable for compounding are optimal fillers for composite production. Their presence of such a filler will allow reducing the amount of polymeric matrix used, allowing for a 10–15% reduction of biopolyester used in production versus a raw polymeric-matrix-based pot. Biocomposite characteristics have been covered in depth in several books and papers [18,19]. The addition of natural or biobased fibres, which have a lower density and biodegradability, makes biocomposites more affordable and lightweight and promotes biodegradability, and in particular, disintegration [14,20]. Thus, biocomposites which are compostable, and even possibly soil biodegradable, are raising interest in agriculture applications since allow for their disposal post use together with crop cuts in composting, while possible lost plastic debris in the soil would be able to biodegrade and be converted into valuable biomass. Biomass sourced from organic by-products, like plant residues, agricultural waste, and forest by-products, stands as a renewable resource with vast potential to meet the growing demand for sustainable materials [21,22,23,24,25]. Among the available biomass by-products, insect exoskeletons are an innovative emerging resource. Indeed, insect proteins have emerged as a sustainable and versatile protein source, with applications mainly in animal feed. They are considered sustainable because of the reduced land use, the possibility to use agrifood waste as a substrate for insect farming, the fast rate of insect raising, and the limited use of energy promoting that generated from renewable resources [26]. Specifically, insects, such as crickets, emerge as noteworthy valuable alternatives protein sources due to their fast and relatively low-cost raising [27]. The insect exoskeletons that remain after the protein extraction procedure with alkaline is about 15–20% by weight, where protein is about 70% of the cricket biomass, plus 10–15% lipids. This is composed of calcium carbonate (80–90%) and chitin (10–20%) and is considered a no-value biomass [28]. Considering either impact allocation by weight or by value, the impact of cricket exoskeletons is extremely low, with most of the impact from crickets growing going to be allocated to the main products, being the insect protein. In the present study, these exoskeletons were dried, milled, and used as filler for biocomposites to be devoted to injection moulding item production. Representative prototypes of biocomposites were produced at industrial scale as plant pots, which confirmed the suitability of insect exoskeleton powder to be used as a filler in biodegradable polymeric matrices, and were positively validated for plant growing, compostability, and soil biodegradability.

## 2. Materials and Methods

### 2.1. Composite Materials

Polyhydroxyalkanoate-co-valerate (PHBV) is characterised by a density of 1.25 g/cm^3^ and melt flow index (190 °C, 2.16 kg) of 15–20 g/10 min. The PBSA FD92PM was from Mitsubishi, with a density of 1.24 g/cm^3^ (23 ± 0.5 °C) [29], melting temperature of 84 °C, melt flow rate (MFR) of 4 g/10 min (ISO 1133 190 ◦C/2.16 kg) [30], and a butylene adipate content of about 20 wt.% [31]. The PBSA FD92PM is certified as industrial/home compostable and biodegradable in soil by TÜV Austria and as food contact grade by EU10/201126 [29]. Cricket insects were provided by the company Nutrinsect (Montecassiano, MC, Italy), with about 85% calcium carbonate and 15% chitin as the main components; the insects were finely ground and subsequently sifted using a 500-micron stainless steel mesh sieve from Gilson, Columbus, OH, USA. The finely ground and sifted powder was then placed in an oven at 60 degrees Celsius for 24 h to eliminate any potential moisture before usage. The thermal stability was evaluated by a thermal gravimetric analysis, as discussed below, and the result was compatible with processing in extrusion with the selected polymeric matrix.

### 2.2. Composite Preparation on Laboratory Scale

Formulations were prepared with varying insect exoskeleton filler (I) content, namely, 0%, 5%, 10%, and 15% by weight relative to the total weight of the composite, as reported in Table 1. These formulations were based on a polymeric matrix with a 70/30 weight ratio of PBSA/PHBV (M). This diverse set of formulations allowed for a comprehensive analysis of the impact of the insect exoskeleton filler content on the properties of the composites, as experienced in previous studies [14,24]. 

Moreover, with the content of filler higher than 15% by weight, the processing by injection moulding at an industrial scale was challenging and the pots produced were defective.

Polymeric blends and composites were prepared by using a laboratory-grade single-screw extruder to produce monofilaments. This specialised extruder, manufactured by Brabender in Duisburg, Germany, boasts a screw with a diameter of 19 mm and a nominal length equivalent to 25 screw diameters. It is seamlessly interfaced with a computer and has four independently adjustable heating zones. This allows for the precise control and regulation of the temperature within different segments of the extruder, ensuring optimal processing conditions for monofilament production. The extruder operating conditions adopted for all the formulations were 175/180/175/160 °C, with the die zone at 150 °C, and the total mass flow rate was 2 kg/h with a screw rate of 100 rpm. The extruded strands underwent cooling in a water bath at ambient temperature and were transformed into pellets using an automated knife cutter (PROCUT 3D, Chinchio Sergio Srl, Tradate (VA), Italy). The pellets were used to produce a dog-bone specimen for a tensile test. The tensile samples were manufactured using a mini-injection press (ZWP Proma, Tychy, Poland). In the production of the dog-bone specimens, the pellets were introduced into the temperature-controlled barrel of the injection press set at 140 °C. After approximately 1 min, the molten material was mechanically injected into the stainless-steel dog-bone mould and held at 60 °C for an additional minute before extracting the sample. All the materials utilised in the process were dried at 60 °C for 48 h before processing to minimise the possible degradation for hydrolysis of the biopolyesters during processing in the extruder.

### 2.3. Composite Characterisations

The thermal stability of the composites obtained was investigated by thermal gravimetric analysis (TGA). These measurements were carried out in duplicate on about 10 mg of the sample by using a Netzsch STA 200 Regulus (Selb, Germany) under nitrogen flow (20 mL/min) at a heating speed of 10 K/min from 30 °C to 700 °C. Differential scanning calorimetry (DSC) was performed by a Perkin Elmer DSC 6000 (Perkin Elmer Instrument, Waltham, MA, USA). About 15 mg of pellets were placed in an aluminium pan and subjected to a first heating from −60 °C to 185 °C (to remove any thermal history from processing), followed by cooling from 185 °C to −60 °C and a second heating to 185 °C under a nitrogen flow (20 mL/min at 10 °C/min). The melting temperature (Tm) and the variation in melting enthalpy (∆*H_m_*) were determined by the second-heating DSC curves. The crystallinity of each polymer was determined from the melting enthalpy in the endotherm (∆*H_m_*) using$$X_{c}\;=\;\frac{{∆H}_{m}}{{w_{i}∆H}_{m}^{0}}\;\times \;100\%$$
where w*_i_* is the weight content of the corresponding polymer and ${∆H}_{m}^{0}$ is the melting enthalpy of a 100% crystalline polymer with pure PHBV (146 J/g) [32] and PBSA (113.4 J/g) [33]. Tensile tests were conducted on the dog-bone specimens (Type V) while adhering to the ASTM D638 standards [34], with dimensions verified to meet the tolerance requirements: a length of 80 mm, a larger section width of 12 mm, a narrower section width of 4 mm, and a thickness of 2 mm. Tensile tests on the samples prepared with the injection moulder were performed at room temperature and a crosshead speed of 10 mm/min using an INSTRON 5500 R universal testing machine (Canton, MA, USA) equipped with a 10 kN load cell and interfaced with a computer running the Merlin software (Canton, MA, USA) MERLIN software (INSTRON version 4.42 S/N–014733H).

The morphology of the insect exoskeleton powder and the biocomposites was investigated by scanning electron microscopy (SEM) using a COXEM Co., Ltd. Model EM-30 N (Daejo, Republic of Korea). The samples were frozen under liquid nitrogen and then fractured along the cross-section to cause a fragile fracture, which provided a smoother surface for the SEM analyses. The sample’s fractured surface was metallised with a thin gold layer using a sputter coater (Edward S150B, Crawley, West Sussex, UK) before the microscopy to avoid charge buildup.

An MCR 92 rheometer from Anton Paar Gmbh, Graz, Austria, equipped with a 25 mm diameter plate-plate configuration, was used to examine the effect of the insect exoskeleton filler content on the viscosity of the PBSA/PHBV composites. Before testing, each sample underwent a drying process and was then melted on a rheometer plate for 3 min at constant temperatures to eliminate any thermal history associated with processing. The measurements were carried out at four different temperatures (150, 160, 170, and 180 °C) using a 1 mm plate–plate configuration. To establish suitable operational parameters within the linear viscoelastic range, an amplitude strain sweep was conducted. Oscillatory angular frequency sweeps (ω) that ranged from 0.314 rad/s to 628 rad/s (0.05 to 100 Hz) were performed using a strain amplitude (γ) of 0.2%. The complex viscosity (η*), loss modulus (G″), and storage modulus (G′) were measured as functions of the angular frequency (ω).

### 2.4. Scale-Up Production of Prototypes

The most effective formulations for the plant pots were prepared and fine-tuned, which yielded the following conclusive compositions:PBSA/PHBV 70/30 (M).PBSA/PHBV 70/30 + 15 wt.% of insect exoskeleton filler (M_15).

The plant pots were produced by injection moulding at an industrial scale by LCI/FEMTO Engineering, San Casciano, in Val di Pesa, Italy. The demonstrators were moulded with a Negri Bossi P75 Injection moulding machine (Cologno Monzese, MI, Italy). The injection machine presents 4 extruder zones with control of the temperature profile. The extruder operating conditions adopted for all the formulations were 160/170/170/170 °C.

### 2.5. Compost Degradation Tests

The developed biocomposite pots based on PBSA/PHBV and PBSA/PHBV/15% insect exoskeleton powder were produced from polymers that were already certified according to EN 13432 [16], with the optional addition of insect exoskeleton material. The insect exoskeleton filler was not chemically modified and did not contain any additional organic additive in a concentration above 1% (dry weight), and the total sum of these organic constituents without a determined biodegradability did not exceed 5%. In this case, disintegration was studiedto attest to the compostability according to EN13432 [16]. The disintegration of the demonstration PBSA/PHBV pot (1.55 mm bottom, 1.56 mm sidewall) and PBSA/PHBV/15% insect exoskeleton powder pot (1.52 mm bottom, 1.60 mm sidewall) was qualitatively tested according to ISO 16929 [35]. The thickness is an important characteristic, as a high thickness will impact the disintegration rate negatively. The pilot-scale aerobic composting test simulated as closely as possible a real and complete composting process in composting bins of 200 L. The pots both had a height of 15 cm and bottom diameter of 14 cm and were cut into pieces of 5 cm × 5 cm. They were added to a mixture of fresh Vegetable, Garden, and Fruit waste (VGF) and structural material and introduced in an insulated composting bin, after which composting spontaneously started. Like in full-scale composting, inoculation and temperature increases happen spontaneously. During the composting, the contents of the vessels were turned manually, at which time the test item was retrieved and visually evaluated. The fresh biowaste was derived from a separately collected organic fraction of municipal solid waste, which was obtained from the waste treatment plant of Erembodegem, Belgium. The biowaste at the start should have a moisture content and a volatile solids content of the total solids (TSs) of more than 50% and a pH above 5. From Table 2, it can be observed that these requirements were fulfilled. The biowaste contained a moisture content of 73.1% and a volatile solids content of 85.3% of the TSs. At the start, a pH of 5.6 was measured, and after 1.7 weeks of composting, the pH had increased to above 8.5. Furthermore, the C/N ratio of the biowaste at the start should preferably be between 20 and 30. An optimal C/N ratio of 28 was found for the biowaste.

Figure 1 shows the temperature evolution during the composting process and the prescribed temperature conditions. Each time the temperature criteria were not fulfilled, action was taken to maintain optimal temperature conditions. Also, during the composting process, the bin was placed in different incubation rooms (at ambient temperature, at 40 °C, and at 45 °C) to maintain optimal temperature conditions during the different time periods.

Elevated temperatures during the composting process were also caused by the turning of the contents of the bin, during which air channels and fungal flakes were broken up, and the moisture, microbiota, and substrate were divided evenly. As such, optimal composting conditions were re-established, which resulted in higher activity and a temperature increase. The temperature profile showed an initial thermophilic phase and a mesophilic continuation, which is representative of industrial composting, and therefore, it can be concluded that the temperature conditions were fulfilled. The oxygen concentration in the exhaust air always remained above 10%. As such, good aerobic conditions were guaranteed.

## 3. Discussion

### 3.1. Thermal Analysis of Composites

The thermal properties of biocomposites can be significantly influenced by the process of melt blending the polymers and incorporating the biomass fillers. To comprehensively evaluate these effects, we conducted a thermogravimetric analysis (TGA), as reported in Figure 2.

The PBSA/PHBV blend and the biocomposites that contained insect exoskeleton fillers at concentrations of 5%, 10%, and 15% by weight exhibited a distinctive two-stage degradation pattern. In this thermal process, the initial peak degradation temperature could be attributed to the PHBV, while the subsequent peak degradation temperature corresponded to the PBSA. These findings are in line with what was observed in previous research [36,37]. Notably, the thermal decomposition of the filler, which in this case consisted of insect exoskeletons, exhibited a distinctive one-step degradation process at about 200 °C.

The analysis revealed specific thermal parameters for the materials: the onset degradation temperature related to the PHBV and the temperature of degradation corresponding to the PBSA were measured at about 190 °C and 250 °C, respectively. Notably, these values exhibited a decreasing trend as the filler loading increased, indicating that the addition of fibres had a diminishing effect on the thermal stability of the biocomposites.

To implement the characterisation of the thermal properties and structure of the materials, additional thermal analyses were performed, which allowed for the determination of the melting and crystallisation characteristics using differential scanning calorimetry (DSC). The second-melting and cooling curves in Figure 3 and Figure 4, respectively, offer further insights into how the polymer blending and the introduction of fibrous materials influenced the thermal properties of these biocomposites. This combined analysis ensured a clear and cohesive understanding of the thermal behaviour of the materials. Differential scanning calorimetry (DSC) analysis was carried out on all the samples, and the results recorded during the second heating are summarised in Table 3.

The PBSA/PHBV blend displayed two major melting phenomena (Figure 3) and two melt crystallisation phenomena, corresponding to the PBSA and PHBV, respectively. PHBV is a semi-crystalline polymer with a glass transition temperature of around 5 °C and a melting temperature of 145–150 °C [33]. The PBSA/PHBV blend displayed double-melting peaks, with the first less intense melting peak (Tm1) at 77.60 °C and the second main melting peak (Tm2) at 87.40 °C. The PHBV displayed a single melting peak at 113 °C. The double-melting behaviour of polymers, such as PBSA, has been widely reported in the literature. It is ascribed to the melt recrystallisation of polymers during the heating process. Imperfect crystals may melt at lower temperatures, recrystallise, and subsequently melt at higher temperatures, leading to double-melting peaks [33,36,37,38]. In general, the melting temperatures of PBSA and PHBV in the blend and the biocomposites containing the insect exoskeleton biomass occurred in the same range, where Tm1 for both the PBSA and PHBV slightly decreased with biomass filler loading.

The PBSA/PHBV blend had two melt recrystallisation peaks, with the first melt recrystallisation peak (Tc1) at 51.33 °C corresponding to the PBSA and the second melt recrystallisation peak (Tc2) at 113 °C corresponding to the PHBV (Figure 4). The melt crystallisation temperatures of the PBSA shifted to lower temperatures with the addition of the filler. This indicates that the crystallisation of PBSA was restricted by the presence of the biomass due to the suppressed nucleation. In contrast, in the case of PHBV, there was an increase in the crystallinity with the addition of the filler load. This behaviour was attributed to the interactions between the polymeric matrix and the filler, which promoted a more ordered arrangement of the polymer chains. As the filler content increased, it acted as a nucleating agent, which encouraged the formation of crystalline regions within the composite.

This enhanced crystallinity positively influenced the mechanical properties and thermal stability of the developed PHBV-based composites. This underscores the intricate relationship between the filler concentration and crystallinity, a crucial consideration in tailoring composite materials for specific applications.

### 3.2. Mechanical and Rheological Analyses of Composites

The results of the tensile tests, shown in Figure 5, reveal that as the filler content within the matrix increased, the mechanical properties remain relatively consistent. The tensile strength (Figure 5a) remained relatively unchanged as the filler load increased within the matrix. For the Young’s modulus (Figure 5b), a slight increase was observed when the fibre load was increased from 5% to 15%; however, this increase did not significantly impact the mechanical properties. The elongation at break (Figure 5c) showed a notable decrease from 5% to 10% of the filler load, with a more drastic reduction occurring at 15%. This suggests that in this scenario, the primary role of the filler was to act as a filler rather than reinforcing the composite, which emphasises the importance of considering filler content when tailoring material properties. The mechanical properties presented by the biocomposites are in line with those presented from previous biocomposites based on a biopolyester matrix with natural fillers, such as bran, which were found to be valuable for pot production and use [39,40,41].

In Figure 6, the PBSA–PHBV matrix between 150–160 °C and 170 °C exhibited overlapping viscosity behaviour, while a distinct change occurred at 180 °C, signifying that fusion truly began at this temperature. Although the material was not fully melted at 150–160 °C and 170 °C, the viscosity did not change as significantly with temperature in these ranges. In the case of sample M_5, the viscosities start to differentiate minimally, with a notable increase in viscosity values at 180 °C. For sample M_10, the viscosities changed with temperature, presenting the lowest value at 180 °C, and the values at 150, 160, and 170 °C were not distinguishable one from the other. Sample M_15 showed an intermediate behaviour between M_5 and M_10, where the values at 150–160 °C were very close, while the samples at 170 and 180 °C were markedly different.

Figure 7 shows the values for composites M_10 and M_15, which were similar, especially in the higher angular frequency range. The pure matrix exhibited a more pronounced shear-thinning behaviour compared with the others, while the M_5 composite showed a lower viscosity value than the other composites up to a range of average angular frequencies, then aligned with the matrix values at high angular frequencies.

At the melting temperature of 180 °C, the storage modulus exhibited a noticeable deviation from the normal behaviour in the modulus curve for both M and M_5 at low angular frequencies. This implies potential material degradation, likely associated with the matrix itself, particularly concerning the degradation of the PHBV. In contrast, for samples M_10 and M_15, this degradation seems to have been mitigated by the reinforcing effect of the filler, which effectively concealed this degradation effect.

### 3.3. Morphological Analysis of Composites

Conducting a morphological analysis is a pivotal step in assessing the efficacy of the insect exoskeleton filler’s dispersion within the polymeric matrix and comprehending the interactions between the matrix and fillers. This analysis involved a detailed examination of the fractured surfaces of the specimens, which provided critical insights into the structure and composition of these composites. The Scanning Electron Microscopy (SEM) images, presented in Figure 8, highlight significant findings that reveal that the analysis of the filler incorporated into the matrix exhibited an uneven morphology. Specifically, it appeared in the form of flakes and short fibres, which suggests a lack of uniformity in its structure. This detailed examination provides valuable insights into the filler’s characteristics within the composite. The surface morphology of the pure matrix (M) appeared exceptionally smooth. This observation held across all the SEM images, irrespective of the filler loading, signifying a consistent and uniform distribution of (I) fillers throughout the thermoplastic matrix. Notably, the biocomposite that contained 15% by weight of filler stood out due to its remarkable dispersion within the matrix by showcasing an exceptionally even distribution.

### 3.4. Disintegration in Compost

The disintegration of the PBSA/PHBV pot (thickness: ±1.55 mm (bottom), ±1.56 mm (sidewall)), cut into 5 cm × 5 cm pieces, is reported in Figure 9.

After one week in the composting process, the samples of test material were abundantly present and mostly intact. At week 3, the material turned very fragile, with cracks and tears. After 4 weeks, even the growth of fungi on the samples’ surfaces was noted; after 8 weeks, a large presence of fungi was observed but the pieces of material remained mainly intact; after 10 weeks, the tears increased, but after 12 weeks, still a lot of large test material pieces could be retrieved. Thus, with the present thickness, the pot would not pass the standard for compostability since it failed the disintegration step. However, by reducing the thickness of the pots, this requirement might still be fulfilled, but the mechanical properties might be compromised. The present items were produced with a standard mould for pots, and thus, with common dimensions for this type of item.

In Figure 10, the visual observation of PBSA/PHBV/15% insect exoskeleton powder pot (thickness: ±1.52 mm bottom, ±1.60 mm sidewall), cut into 5 cm × 5 cm pieces, is reported. After one week in the composting process, the samples of test material were abundantly present and mostly intact. At week 3, the material turned very fragile, and at 4 weeks, the growth of fungi was noted, but the material was still mostly intact. After 8 weeks, a large presence of fungi was observed and the test material had mostly fallen apart into pieces of variable sizes. After 10 weeks, the material had completely fallen apart into pieces of variable sizes. At 12 weeks, only a few small pieces of test item could be retrieved. Thus, with the presence of the biomass in the composite, the pots with the current thickness easily met the requirements of the standard for compostability in terms of disintegration. This demonstrated the importance of adding natural fillers in polymeric matrices, even when these were based on biopolyesters since the thickness of the items was a restraint for meeting the requirements of the standard for compostability [42].

Thus, to meet the requirements for the standard for compostability, addressing disintegration, the presence of the natural filler was very relevant. Additionally, the compost extract from both reactors of the PBSA/PHBV and PBSA/PHBV/15% insect exoskeleton powder presented no negative effect on seed germination tested using garden cress (*Lepidum sativum*) seeds. Indeed, PBSA and PHBV were tested in agriculture applications with no phytotoxic effects [43,44,45] and insect exoskeletons, being a natural biomass, are normally present in the environment, and thus, were not expected to generate a negative effect when decomposing in compost or soil.

## 4. Conclusions

Insect farming as a novel source for protein production for animal feed is an economic, sustainable, and natural approach. As for any main production, by-products and wastes are generated, in particular insect exoskeletons that remain after protein extraction. This new and interesting biomass with no value requires valuable applications and was positively tested as a filler in biocomposite production, with particular reference to possible applications in the production of agriculture-oriented items. Various prototype demonstrators, which are designed to enhance the value of byproducts within the framework of a circular economy approach, have been manufactured on an industrial scale.

The materials produced presented valuable thermal, mechanical, and compostability properties that matched the requirements for applications, such as agriculture rigid items (in the present context as plant pots).

Extrusion processing, which was conducted at both laboratory and industrial scales, affirmed the suitability of insect exoskeleton powder as a filler in biodegradable polymeric matrices. This included maintaining the thermal stability, morphological integrity, and adequate mechanical properties necessary for the industrial production of demonstrators, such as plant pots, with requirements compatible for use in green houses and open fields. The production of the pots proceeded seamlessly and aligned with industrial production cycles.

At the end of the disintegration test in the compost for the polymeric matrix with no filler, after 3 months, still quite a lot of large test item pieces were retrieved, and therefore, it was concluded that the disintegration was insufficient for items of the present thickness. Meanwhile, when the insect exoskeleton filler was present, at the end of the disintegration test in the compost, only a few small pieces were retrieved, which meant the material passed the disintegration test requirements. Thus, the presence of the filler had a relevant effect of promoting the material degradability and was necessary to meet the standard for compostability. The pots are currently being tested on the plant growth of peppers plants, and further studies will be conducted to deepen the effect of pot disintegration on compost quality.

## Figures and Tables

**Figure 1 polymers-17-00943-f001:**
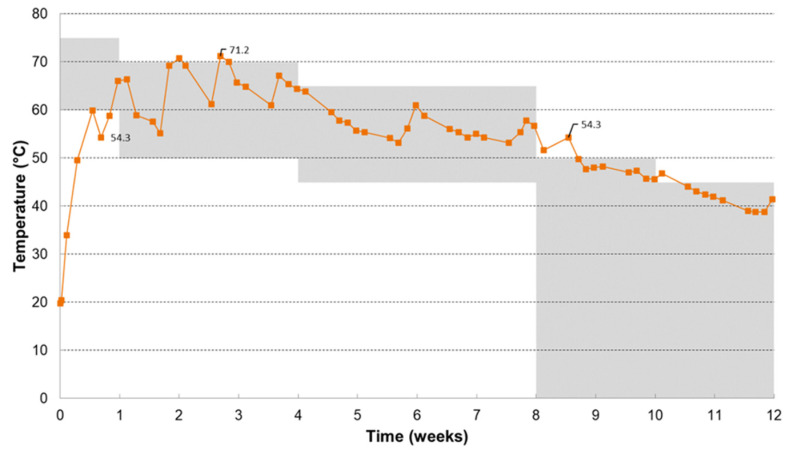
Temperature evolution during the composting test.

**Figure 2 polymers-17-00943-f002:**
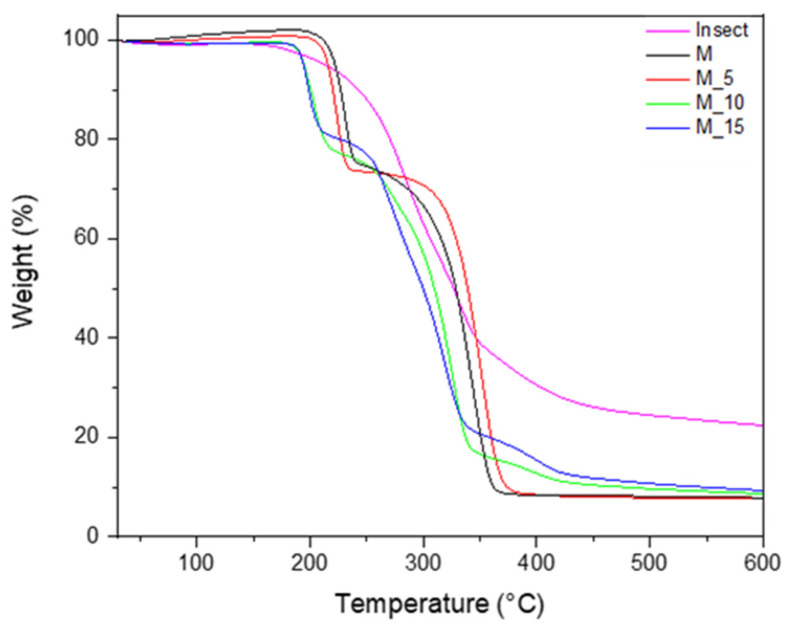
Thermal gravimetric analysis of raw materials (I, M) and composites (M5, M10, M15).

**Figure 3 polymers-17-00943-f003:**
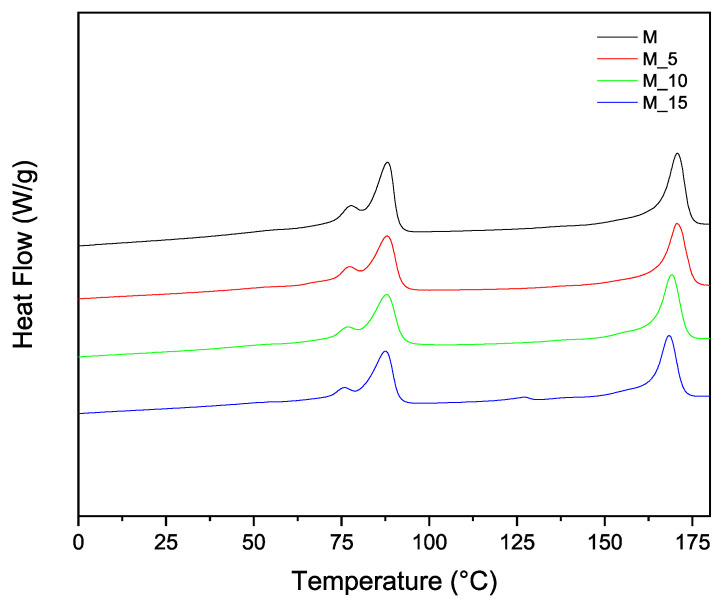
DSC thermograms of polymeric Matrix (M) and composites (M5, M10, M15) (heating run).

**Figure 4 polymers-17-00943-f004:**
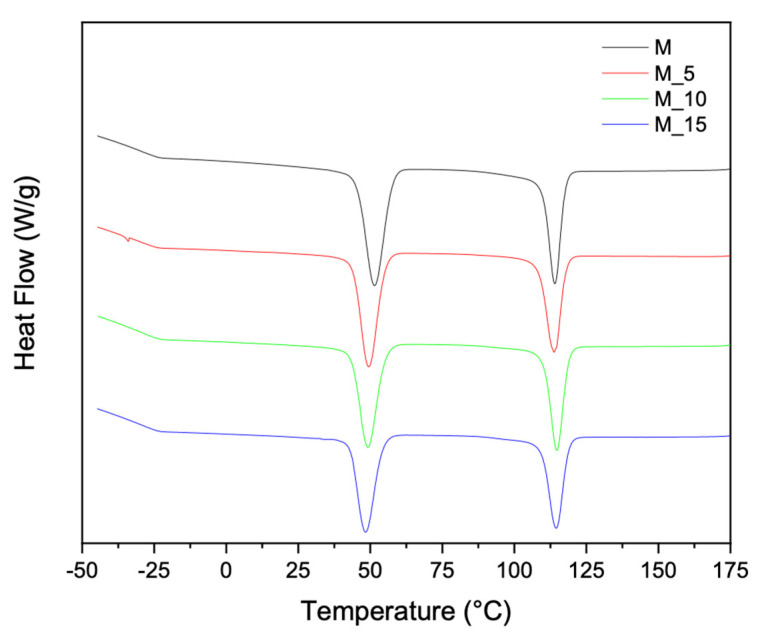
DSC thermograms of polymeric matrix and composites composites (cooling run).

**Figure 5 polymers-17-00943-f005:**
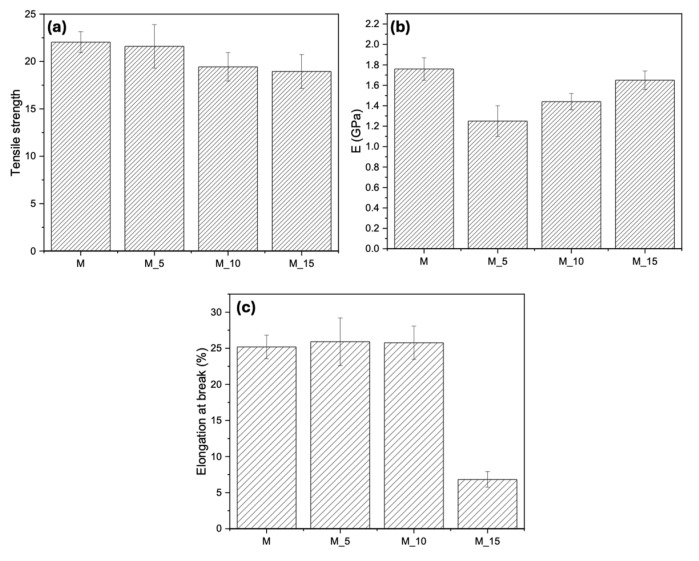
Tensile test results of all the composites developed, (**a**) tensile strength, (**b**) Elastic Modulus, (**c**) Elongation at break.

**Figure 6 polymers-17-00943-f006:**
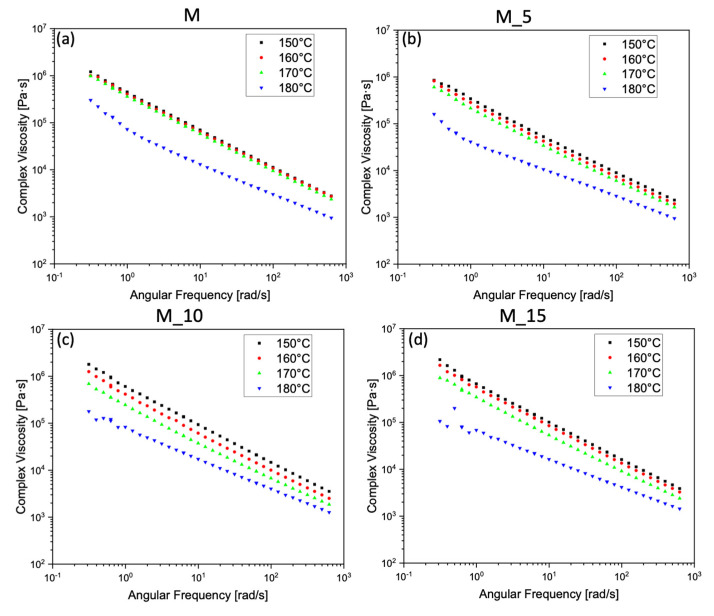
Complex viscosity versus angular frequency in a range of temperatures from 150 °C to 180 °C of: (**a**) the matrix (M), the composites (**b**) M_5, (**c**) M_10, (**d**) M_15.

**Figure 7 polymers-17-00943-f007:**
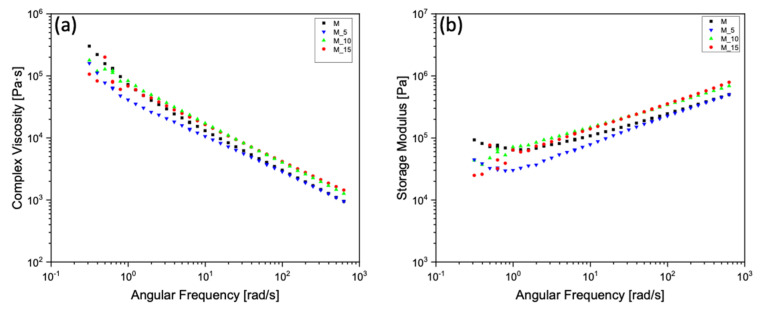
Complex viscosities as a function of angular frequency at 180°C (**a**) for the matrix (M) and the composites M_5, M_10, M_15, and storage moduli (**b**) of the matrix (M) and the composites M_5, M_10, M_15 at 180°C.

**Figure 8 polymers-17-00943-f008:**
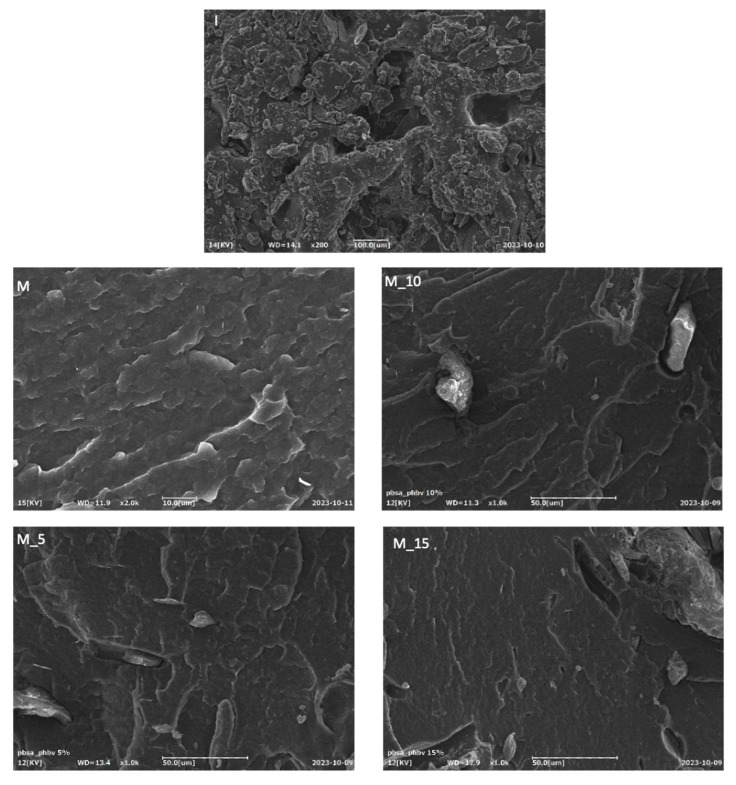
Morphologies of pure insect exoskeleton filler (I) and PBSA/PHBV-based composites (M) with 5% (M5), 10% (M10) and 15% (M15) by weight of filler in the polymeric matrix.

**Figure 9 polymers-17-00943-f009:**
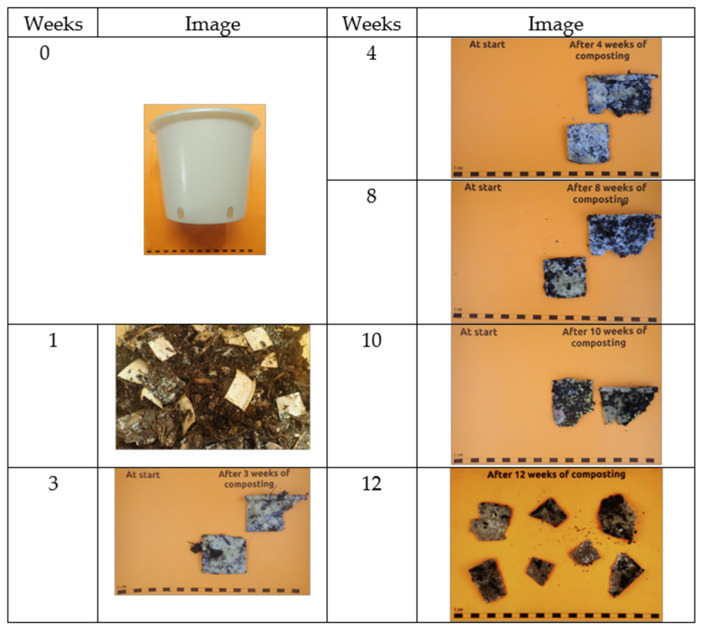
Visual observations of a PBSA/PHBV pot, cut into 5 cm × 5 cm pieces, during the composting process.

**Figure 10 polymers-17-00943-f010:**
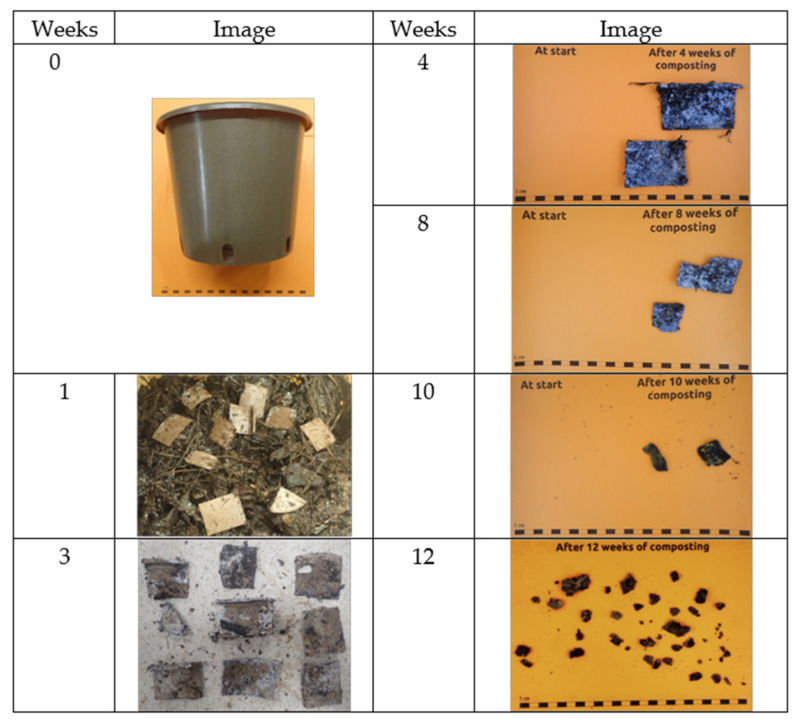
Visual observations of the PBSA/PHBV/15% insect exoskeleton powder pot, cut into 5 cm × 5 cm pieces, during the composting process.

**Table 1 polymers-17-00943-t001:** Label of polymeric matrix (M) and biocomposite samples.

Name	Insect Exoskeleton Content (wt-%)
M	0
M_5	5
M_10	10
M_15	15

**Table 2 polymers-17-00943-t002:** Characteristics of the biowaste at the start of the composting.

Characteristics	Biowaste
Total solids (TSs, %)	26.9
Moisture content (%)	73.1
Volatile solids (VSs, % of TSs)	85.3
Ash content (% of TSs)	14.7
pH	5.6
Electrical conductivity (EC, µS/cm)	3990
Volatile fatty acids (VFAs, g/L)	2.6
NOx--N (mg/L)	<10.0
NH4+-N (mg/L)	528
Total N (g/kg TS)	15.4
C/N	28

**Table 3 polymers-17-00943-t003:** Thermal data from the DSC analysis of the investigated samples.

Sample	Tm (°C)		Tc (°C)		Xc (%)	
	PBSA	PHBV	PBSA	PHBV	PBSA	PHBV
M	88.2	170.6	51.5	113.1	49.8	60
M_5	88	170.5	49.5	113.8	44.8	63.6
M_10	87.8	169.1	49.2	114.7	42.6	66.2
M_15	87.5	168.3	48.3	114.5	41	67.3

## Data Availability

Data available upon request.

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
