# Peer review of "Valorising Insect Exoskeleton Biomass Filler in Bioplastic-Based Eco-Friendly Rigid Items for Agriculture Applications"

_polymers, 2025, doi:10.3390/polym17070943_

Round 1
Reviewer 1 Report
Comments and Suggestions for Authors
The manuscript presents a promising approach to sustainable biocomposites for agriculture, successfully integrating industrial scalability and compostability. However, addressing the mechanical trade-offs, providing economic/sustainability metrics, and clarifying unresolved limitations (e.g., thickness effects) would significantly enhance its impact. With revisions, this work could serve as a valuable contribution to the field of biodegradable materials. The comments are shown below:
- The unfilled PBSA/PHBV matrix failed disintegration tests due to thickness. While the authors suggest reducing thickness, this critical limitation should be addressed with additional data (e.g., testing thinner samples) to validate the claim.
- The significant reduction in elongation at break (up to 15% filler) raises concerns about brittleness. A discussion on balancing filler content with mechanical requirements for agricultural applications (e.g., durability during handling) is needed.
- The paper asserts reduced production costs but lacks quantitative data (e.g., cost/kg comparison of composites vs. pure polymers). Including a cost-benefit analysis would strengthen the argument.
- The sustainability of insect farming (e.g., energy/water use, scalability) is not discussed. Clarifying the environmental footprint of insect biomass production would enhance the study’s credibility.
- For agricultural applications, the ecological effects of degraded filler (e.g., nutrient release, microbial activity) should be evaluated to ensure no adverse impacts.
- Figures referenced in the text (e.g., Figures 2–10) are missing, limiting the ability to assess data quality. Ensure all figures are included and labeled clearly.
- Compare the performance of insect filler with other low-cost fillers (e.g., wood flour, rice husk) to highlight its unique advantages.
- Conclusions: Emphasize the novelty and suggest future directions. In addition, it is advised that the refined summaries should be better presented point by point.
- Propose follow-up studies on long-term field trials for biodegradation and plant growth performance in soil.
- The format of references should be checked, especially the consistency.
Author Response
Answers to Reviewers 1
Dear Editor, dear Reviewer,
first of all we want to thank you for the review of our manuscript and the most appreciated invitation to submit a revised version. The comments provided are invaluable in order to significantly improve the quality of the manuscript and clarify some aspects of the paper that needed to be implemented. Therefore we addressed each single comment as mentioned below.
Reviewer 1
The manuscript presents a promising approach to sustainable biocomposites for agriculture, successfully integrating industrial scalability and compostability. However, addressing the mechanical trade-offs, providing economic/sustainability metrics, and clarifying unresolved limitations (e.g., thickness effects) would significantly enhance its impact. With revisions, this work could serve as a valuable contribution to the field of biodegradable materials. The comments are shown below:
- The unfilled PBSA/PHBV matrix failed disintegration tests due to thickness. While the authors suggest reducing thickness, this critical limitation should be addressed with additional data (e.g., testing thinner samples) to validate the claim.
The claim has been modified, stressing the importance of natural fillers to meet disintegration requirements as an option to be preferred to raw polymer pots (or agriculture rigid items in general) since the author had access to a standard mould for pots in an industrial facility, and indeed, we thank the referee for outlining this, we cannot guarantee that reducing the thickness the mechanical properties for pots would be maintained. While the positive tests made in University Miguel Hernandez with these pots on peppers plants confirm that they are plenty usable in real conditions, but this study is to be published by this agronomic expert research team.
- The significant reduction in elongation at break (up to 15% filler) raises concerns about brittleness. A discussion on balancing filler content with mechanical requirements for agricultural applications (e.g., durability during handling) is needed.
The authors have introduced references to previous work where they tested pots with similar mechanical properties in real plant growing situation and obtained valuable results in performance. Thus indeed the present is a filler and not a strengthening fiber, with purpose of cost reduction and promotion of biodegradation, but the reduction in mechanical properties is still in the range of usability for pots in plant growth.
- The paper asserts reduced production costs but lacks quantitative data (e.g., cost/kg comparison of composites vs. pure polymers). Including a cost-benefit analysis would strengthen the argument.
We added an estimation of the reduction cost for the pots production, the insect filler is estimated as no value at all from the producer (Nutrinsect) which at present has an issue for its disposal, the presence of the filler allows a reduction in use of the expensive polymers (PBSA/PHBV) of 10 to 15%.
- The sustainability of insect farming (e.g., energy/water use, scalability) is not discussed. Clarifying the environmental footprint of insect biomass production would enhance the study’s credibility.
Thanks for the comment, we added reference to a study on insect farming footprint that is considered significantly convenient versus other sources of protein for animal feed.
- For agricultural applications, the ecological effects of degraded filler (e.g., nutrient release, microbial activity) should be evaluated to ensure no adverse impacts.
This effect was validated by University Miguel Hernandez, and will be published in an incoming paper,
- Figures referenced in the text (e.g., Figures 2–10) are missing, limiting the ability to assess data quality. Ensure all figures are included and labeled clearly.
We are sorry for the figures being not displayed in the manuscript, we checked this in the manuscript.
- Compare the performance of insect filler with other low-cost fillers (e.g., wood flour, rice husk) to highlight its unique advantages.
The present filler is mainly based on calcium carbonate and chitin, we added comparison with composites that were prepared with paper sludge, very similar being calcium carbonate and cellulose.
- Conclusions: Emphasize the novelty and suggest future directions. In addition, it is advised that the refined summaries should be better presented point by point.
We thank the referee for his suggestion and reflected it in the manuscript.
- Propose follow-up studies on long-term field trials for biodegradation and plant growth performance in soil.
Those test are currently running, Organic Waste System has confirmed biodegradability even in soil (and marine water) for the pots with biomass fillers, while University Miguel Hernandez has confirmed positive growing of pepper plants in the pots, these data will be reported in a follow up paper. We mentioned this in the conclusions.
- The format of references should be checked, especially the consistency.
We apologise for the confusion in references, we checked the format and coherence, and implemented them.

Reviewer 2 Report
Comments and Suggestions for Authors
Paper tackles an interesting issue and can be published after its minor review. Methods are well known and the only novelty is related to the material used (exoskeletons of insects), thus it is more a data report than a scientific original work.
Comments:
Title: should be more scientific/precise, less poetic
Abstract: please place numbers (filler %) and more details; precise conclusion from the research
Line 41, 67: double spacing
Line 76: references needed
Insect’s exoskeletones or insects’ ?
More information about the chemical composition and physical properties of insects’ exoskeletons should be placed in the introduction
Detailed aims of the paper and hypothesis are not listed in the introduction
No “weak points” or drawbacks of the polymer matrix are mentioned making the introduction more like a publicity not an overview
Why 5-10-15 % filler content was chosen? How does this refer to similar works? Or maybe some theoretical considerations?
Have You obtained composites or blends?
Some norms and standards should be better described for a reader (not only by number); just one sentence to make even those not working in the field understanding what was done
Author Response
Answers to Reviewer 2
Dear editor and dear Reviewer,
first of all we want to thank you for the review of our manuscript and the most appreciated invitation to submit a revised version. The comments provided are invaluable in order to significantly improve the quality of the manuscript and clarify some aspects of the paper that needed to be implemented. Therefore we addressed each single comment as mentioned below.
Paper tackles an interesting issue and can be published after its minor review. Methods are well known and the only novelty is related to the material used (exoskeletons of insects), thus it is more a data report than a scientific original work.
The authors have better stressed the innovation related to exoskeleton biomass and the relevant effect on meeting compost ability standard just when these bioffiller are present.
Comments:
Title: should be more scientific/precise, less poetic
Title has been modified accordingly
Abstract: please place numbers (filler %) and more details; precise conclusion from the research
We have modified the abstract accordingly
Line 41, 67: double spacing
We corrected the spacing
Line 76: references needed
Done
Insect’s exoskeletones or insects’ ?
The material used was powdered insect’s exoskeleton
More information about the chemical composition and physical properties of insects’ exoskeletons should be placed in the introduction
We added more details on this, physical properties info were added in materials section.
Detailed aims of the paper and hypothesis are not listed in the introduction
We improved this part in the introduction
No “weak points” or drawbacks of the polymer matrix are mentioned making the introduction more like a publicity not an overview
We better evidenced limits of cost, and risk of failing the certification for being compostable that would be the main plus of these materials in agriculture applications versus petro polymers not compostable and hardly recyclable in these applications
Why 5-10-15 % filler content was chosen? How does this refer to similar works? Or maybe some theoretical considerations?
Previous experience on these type of composites suggested these ratio, this was evidenced in the materials and discussion, as well as we specified that with higher amount there were issues in processing on industrial scale.
Have You obtained composites or blends?
We refer to blends when just PBSA and PHBV are blended in the polymeric matrix, and compsites when the fillers are present, we checked and amended the text
Some norms and standards should be better described for a reader (not only by number); just one sentence to make even those not working in the field understanding what was done
We adjusted the citation, thanks for evidencing this.

Round 2
Reviewer 1 Report
Comments and Suggestions for Authors
For this submission, the comments are well addressed in association with the improved manuscript in terms of the quality control. A tip for the authors is that, it is important to have some key revisions with the responses to comments, for better capturing the improvements. Whatever, I still recommend accepting this manuscript after minor self-checks.